# Dietary Lipids, Gut Microbiota, and Their Metabolites: Insights from Recent Studies

**DOI:** 10.3390/nu17040639

**Published:** 2025-02-11

**Authors:** Farzad Mohammadi, Iwona Rudkowska

**Affiliations:** 1Endocrinology and Nephrology Unit, CHU de Québec—Université Laval Research Center, 2705 Laurier Blvd, Québec, QC G1V 4G2, Canada; farzad.mohammadi.1@ulaval.ca; 2Department of Kinesiology, Faculty of Medicine, Université Laval, Québec, QC G1V 0A6, Canada

**Keywords:** fatty acids, sterols, gut microbiome, short-chain fatty acids, bile acids

## Abstract

Dietary lipid intake can influence the gut microbiota (GM) and their metabolites, such as short-chain fatty acids (SCFA) and bile acids, which are key mediators of health. The objective is to examine how dietary lipids’ quantity and quality influence the GM and metabolite profiles. A literature review of 33 studies in animals and humans was performed on the effects of saturated fatty acids (SFAs), monounsaturated fatty acids (MUFAs), polyunsaturated fatty acids (PUFAs), *trans*-fatty acids (TFAs), and sterols on GM composition and gut-derived metabolites. The results show that diets rich in MUFAs, n-3 PUFAs, and short-chain FAs have the potential to enhance beneficial bacteria and metabolites. In addition, *trans*-palmitoleic acid, conjugated linoleic acid, and phytosterols may also have potentially beneficial effects on GM, but more research is needed. Medium-chain FAs and n-6 PUFAs have variable effects on the GM. Conversely, intakes of high-fat diets, long-chain SFAs, industrial TFAs, and cholesterol disrupt GM balance. In conclusion, animal studies clearly demonstrate that dietary fats influence the GM and related metabolites. Yet, human studies are limited. Therefore, well-designed human studies that consider the whole diet and baseline health status are needed to better understand the effects of dietary lipids on GM.

## 1. Introduction

A daily intake of total fat of around 20–35% of total energy intake has been recommended for a healthy diet [1,2]. Specifically, dietary guidelines have recommended limiting saturated fatty acids (SFA) to less than 10% of total energy intake; achieving *trans*-fatty acid (TFA) intake to below 1% of total energy intake; and aiming for *omega*-6 polyunsaturated fatty acids (n-6 PUFA) to contribute to 2.5–9% of total energy intake, with *omega*-3 polyunsaturated fatty acids (n-3 PUFA) accounting for 0.5–2% of total energy intake and dietary cholesterol intake to be as low as possible [1]. Dietary fats are absorbed in the small intestine to provide energy and influence the signaling of satiety hormones [3,4,5,6]. Although fatty acids (FA) and sterols are mainly absorbed in the small intestine, a portion of them can pass through the gastrointestinal tract and potentially affect the microbiome composition of the large intestine. The changes in the composition of the microbiome are influenced by factors such as carbon chain length, saturation, and double-bond position of the dietary fatty acids consumed [7]. Therefore, each type of lipid may modify the microbiome [8,9].

The gut microbiota (GM) is dominated by phyla including *Actinobacteria*, *Proteobacteria*, *Fusobacteria*, *Verrucomicrobia*, *Firmicutes*, and *Bacteroidetes* [10], with the two phyla *Firmicutes* and *Bacteroidetes* accounting for 90% of the GM. In general, beneficial bacteria, such as *Lactobacillus* and *Bifidobacterium* (*Firmicutes* and *Actinobacteria*), promote gut health through the production of short-chain fatty acids (SCFAs) and the modulation of immune responses, while potentially harmful bacteria, such as certain *Proteobacteria* like *Escherichia coli* and *Desulfovibrio*, can contribute to inflammation and metabolic disorders when present in high abundance [11]. The relationship between GM and health is mediated through the interaction between microbial metabolites and the host metabolism [12]. Further, bacterial metabolites are produced because of the breakdown and fermentation of dietary nutrients including fats by gut bacteria [13]. Gut-derived metabolites include SCFAs, bile acids, amino acids, polyamines, vitamins, and secondary metabolites like indoles, phenols, and trimethylamine N-oxide (TMAO). The main SCFAs generated by gut bacteria are acetate, propionate, and butyrate, depending on the type of bacteria. Specifically, Bacteroides species predominantly produce acetate and propionate. *Faecalibacterium prausnitzii* produces butyrate, and *Lactobacillus* species produce acetate [14]. Additionally, branched SCFAs, such as isobutyric, isocaproic, and isovaleric acid, can also be generated by *Clostridium* and *Propionibacterium* species [15]. Moreover, SCFAs can also be consumed directly through dietary sources, such as fermented foods, beverages, and specific supplements, which further improve gut health. Bile acid metabolites are produced by gut bacteria such as *Bacteroides* and *Clostridium* [16]. Together, these microbiomes and metabolites contribute to gut health [17].

The main objective of this review is to investigate how dietary lipid intake, including FAs and sterols, modify gut microbiomes and metabolites. Specifically, this review will examine the impact of fat quantity and quality of FAs and sterols on the GM, as well as the metabolome profiles, to identify diet strategies for optimal gut health.

## 2. Materials and Methods

This narrative review synthesizes the existing literature to explore the impacts of dietary fats, fatty acids, and sterols on the GM and metabolites. Studies were identified through a systematic search of the PubMed, Medline, Scopus, and Google Scholar databases up to April 2024. The search included the following keywords: “dietary fatty acids”, “dietary fat”, “gut microbiota”, “short-chain fatty acids”, “metabolic syndrome”, “gut metabolites”, “long-chain fatty acids”, “saturated fatty acids”, “monounsaturated fatty acids”, “polyunsaturated fatty acids”, “omega-3 fatty acids”, “omega-6 fatty acids”, “trans fatty acids”, “gut dysbiosis”, “microbial diversity”, “gut inflammation”, “host-microbiome interactions”, “cholesterol”, “dietary cholesterol”, “sterols”, “phytosterols”, and “plant sterols”. The inclusion criteria encompassed clinical and pre-clinical studies focused on dietary fats, including FAs and sterols, and their effects on the GM and metabolites, providing a comprehensive overview for this explanatory review. Articles not published in English or lacking full-text availability were excluded. The screening process involved an initial review of titles and abstracts, followed by a full-text review for eligible studies. A total of 33 studies published up to 2024 were selected for inclusion.

## 3. Dietary Fats and Sterols on Microbiome and Metabolite Profiles

### 3.1. Quantity of Fat

Studies have compared the impact of fat quantity, defined as the proportion of dietary fat contributing to total caloric intake, on GM composition and function. Numerous animal studies have indicated that high-fat diet (HFD) consumption can lead to reductions in beneficial microbiomes, such as *Bifidobacterium* and *Faecalibacterium*, together with reductions in the concentrations of fecal SCFAs [18,19,20]. Similarly, a human study demonstrated that a low-fat diet (LFD) (20% fat) for 6 months had a greater α-diversity, together with higher levels of *Faecalibacterium*, while the higher-fat diet (HFD) (40% fat) resulted in higher levels of Bacteroides, together with a lower level of *Faecalibacterium*, in healthy adults [21]. Moreover, the HFD resulted in an increase in fecal concentrations of metabolites, including indole and indoleacetic acid and p-cresol, together with a decreased concentration of SCFA (butyric acid and valeric acid), as well as 3-indolepropionic acid compared to LFD [21]. Overall, an LFD may enhance microbial diversity and beneficial bacteria, while an HFD reduces diversity and increases pro-inflammatory bacteria and metabolites. However, the quality of dietary fat also plays an important role.

### 3.2. Quality of Fat

Previous studies have shown that the GM of mice can differ when diets have equivalent amounts of fat but are obtained from distinct sources, with fat quality referring to the type and composition of FA [22,23]. A mice study showed that the GM of mice fed a control and n-3 PUFA (34% fat of kcal)) for 8 weeks exhibited a distinct composition compared to SFA (34% fat of kcal) and n-6 PUFA (31% fat kcal) [22]. Conversely, the GM of mice fed an HFD with SFA (34% fat of kcal) and n-6 PUFA (31% fat kcal) were similar in composition [22]. Another mice study showed that the abundances of beneficial bacteria such as *Akkermansia muciniphila*, *Lactobacillus*, and *Bifidobacterium* were lower in the GM of mice consuming the lard diet (mainly SFA (45% kcal as fat)) compared to an isocaloric HFD (45% kcal as fat rich in fish oil (n-3 PUFA)) [24]. Similarly, rats who received corn oil (6 mL/kg) over a period of 190 days resulted in increased levels of *Bacteroides fragilis*, *Parabacteroides distasonis*, *Escherichia coli*, *Acidovorax hebreus*, *Clostridium* botulinum, Bacteroides *thetaiotaomicron*, and Bacteroides *uniformis*. In contrast, a fish oil (n-3 PUFA, 6 mL/kg) diet led to a decrease in *E. coli*, *Bacteroides* spp., and Clostridium, while increasing the abundance of *Lactobacillales* spp., which typically results in a healthier microbiome composition [25]. In another study, mice were fed diets containing flaxseed oil (n-3 PUFA), soybean oil (n-6 PUFA), a mixture of soybean and fully hydrogenated soybean oil (SFA), or a standard chow diet (n-3 PUFA, n-6 PUFA, and SFA). The results demonstrated that a diet high in SFA led to greater decreases in the Bacteroidetes-to-Firmicutes ratio compared to either the flaxseed or soybean oil (high PUFA). Specifically, the n-6 PUFA diet resulted in a decrease in *Porphyromonadaceae*, while the SFA diet showed a decrease in *Lachnospiraceae*, relative to pre-diet levels [26]. Moreover, the intake of PUFA-rich diets, including a blend of corn/safflower oil (69% linoleic acid) or a blend of flax/safflower oil (38% linoleic acid and 32% alpha-linolenic acid) increased *Isobaculum*, which is beneficial. Further, an animal study showed that palm oil (high SFA) resulted in the highest levels of acetic acid, followed by rapeseed oil (high MUFA) intake, while lard oil (high SFA) and sunflower oil (high n-6 PUFA) intake had the lowest levels. Animals who consumed linseed oil (high n-3 PUFA) exhibited a higher content of beneficial isovaleric acid compared to palm oil, lard oil, and rapeseed oil intakes. Finally, lard intake had a lower concentration of valeric acid compared to the rapeseed oil (high MUFA), sunflower oil (high n-6 PUFA), and linseed oil (high n-3 PUFA) groups, which were exposed to the harmful effects of SFA [27]. Another human study reported that palm oil (high SFA) resulted in distinct effects on the GM and metabolite profiles compared to rapeseed oil (high MUFA) and sunflower oil (high n-6 PUFA). Specifically, the intake of linseed oil (high n-3 PUFA) increased the beneficial *Akkermansia muciniphila* and SCFA concentrations. In conclusion, diets high in SFA are associated with less favorable gut microbiome composition and metabolite profile effects compared to diets high in MUFA or PUFA [28].

### 3.3. Saturated Fatty Acids (SFA)

SFAs are solid at room temperature due to the absence of double bonds between carbon atoms, which makes them more stable than unsaturated fats [29]. SFAs are commonly found in animal products, such as meat, dairy, and eggs. Plant-based foods like coconut oil and palm oil also contain SFAs [30]. A clinical study demonstrated that dietary interventions, such as high or low SFA from animals or plants, cause modest changes in the GM and that the effects of interindividual differences, such as age, sex, ethnicity, and BMI, outweighed the effects of the diets [31]. Further, SFAs may have varying chain lengths of carbon atoms, including very-long-chain saturated fatty acids (VLCSFA), long-chain saturated fatty acids (LCSFA), medium-chain fatty acids (MCFA), and short-chain fatty acids (SCFA). Yet, studies have mainly investigated the effects of LCSFA (palmitic acid) on the GM.

### 3.4. Long-Chain Saturated Fatty Acids (LCSFA)

Sources of LCSFAs, with carbon atoms from 14 to 18, include dairy fat, tallow, lard, and palm oil [32]. A C57BL/6J mice study with a diet enriched with palm oil (10% of energy) demonstrated an increase in the abundance of Bacteroidetes, while a decrease in the abundance of *Bacilli* and the *Clostridium clusters* XI, XVII, and XVIII (*Firmicutes*) after 3 weeks occurred [33]. An increase in Bacteroidetes may lead to higher levels of acetate and propionate [34], while a decrease in *Firmicutes* might result in a reduction in butyrate levels [35]. In another study, C57BL/6J mice received both an HFD containing 34% SFAs (type of fat not specified) and an HFD enriched with *omega*-6 fatty acids (HFD-n6). The HFD with SFA increased the abundance of *Lactobacillus*, *Erysipelotrichaceae*, *Lachnospiraceae*, and *Pseudoflavonifractor* (*Firmicutes*) after 8 weeks. In contrast, *Allobaculum* (*Firmicutes*), *Barnesiella* (*Bacteroidetes*), *Mucispirillum* (*Deferribacteres*), *Bacteroides* (*Bacteroidetes*), and *Bifidobacterium* (*Actinobacteria*) exhibited a decrease in abundance following the SFA-enriched HFD [22]. Further, there was an increase in pathogenic bacteria, such as *Alistipes* sp. *Marseille-P5997* and *Alistipes* sp. *5CPEGH6*, and a decrease in the *probiotic Parabacteroides distasonis* in mice fed a high SFA diet (43.1% from lard). Additionally, the high SFA diet increased metabolites such as lysophosphatidylcholine (LPC) and lysophosphatidic acid (LPA), which are involved in inflammation, cell signaling, vascular development, and cancer progression [36]. Similarly, an HFD rich in 45% of either palm oil (high LCSFAs), olive oil (high MUFA), safflower oil (high n-6 PUFA), or flaxseed/fish oil (high n-3 PUFA) was administered to C57BL/6J mice for 16 weeks. The palm oil-enriched diet increased the abundance of *Coprococcus*, *Erysipelotrichaceae*, and *Lachnospiraceae* (*Firmicutes*) [37]. Conversely, the abundance of *Bacteroides* and *Bacteroidaceae* (*Bacteroidetes*), as well as *Deferribacteres*, *Actinobacteria*, and *Proteobacteria*, decreased after receiving palm oil [37]. Overall, these findings suggest that LCSFA content may lead to distinct shifts in microbiota composition and metabolite profiles, potentially contributing to inflammatory and metabolic dysregulation.

### 3.5. Medium-Chain Saturated Fatty Acids (MCSFAs)

MCSFAs such as caprylic acid (C8:0), capric acid (C10:0), and lauric acid (C12:0), ranging from 6 to 12 carbon atoms, are naturally present in coconut and palm kernel oils [32]. Coconut oil is especially categorized as an SFA source, comprising approximately 92%, and its MCSFAs make up around 64%, with lauric acid being the predominant component [38,39]. In a 14-day study in pigs, adding 0.3% commercial coconut oil to the feed positively influenced the rectal microbial diversity [39]. The supplementation increased the abundance of beneficial bacteria, such as *Lactobacillus* and *Bifidobacterium*, while specific genera, such as *Fusicatenibacter* and *Mitsuokella*, decreased compared to the group fed the pure commercial diet [40]. Moreover, an 8-week study in Black Sea bream showed 0.1% and 0.8% lauric acid resulted in increases in *Firmicutes*, *Betaproteobacteri*, Gammaproteobacteria, and Clostridia, as well as *Clostridiaceae* [41]. In a 42-day study with broiler chickens, lauric acid supplementation at doses of 500 mg/kg and 1000 mg/kg decreased *Phascolarctobacterium*, Christensenellaceae_R-7_group, and Bacteroides, together with increased Faecalibacterium and Ruminococcaceae_UCG-014 [42]. Yet, the consumption of 25 mL/day of coconut oil as part of an energy-restricted diet for 9 weeks did not result in changes in gut microbial diversity among overweight women [43]. Overall, MCSFAs demonstrate variable effects in animal studies. However, a human study suggests no impact on gut microbial diversity.

### 3.6. Short-Chain Fatty Acids (SCFAs)

Short-chain FAs can be obtained through dietary sources, such as fermented foods and beverages [44]. In addition, supplements of short-chain FA exist, such as acetate (C2), propionate (C3), and butyrate (C4) [45]. A mice study with acetate supplementation (with 200 mmol/L magnesium acetate) for 3 weeks reduced the Firmicutes-to-Bacteroidetes ratio, thereby, ameliorating gut dysbiosis. Additionally, acetate supplementation promotes the proliferation of acetate-producing bacteria and those belonging to the genus Bacteroides *acidifaciens* [46]. In another study, butyrate supplementation (0.5 g/kg/day) increased the amounts of fecal Bacteroidetes and reduced the ratio of Firmicutes/Bacteroidetes in mice after 7 weeks [47]. In C57BL/6J mice, sodium butyrate supplementation (400 mg/kg) with HFD for 16 weeks led to an increase in the relative abundance of *Verrucomicrobia* and a decrease in the abundance of Firmicutes [48]. At the family level, after sodium butyrate supplementation, there was an increase in the relative abundances of *Verrucomicrobiaceae*, *Bacteroidaceae*, and *Alcaligenaceae*, while there was a decrease in the relative abundances of Lachnospiraceae and *Paraprevotellaceae* [48]. In a human study, type 1 diabetes patients who received acetate and butyrate (40 g/day) supplements for 6 weeks demonstrated higher levels of Bacteroides *uniformis*, unclassified Parabacteroides, and *P. distasonis*, with a reduction in levels of *Eubacterium ramulus*, *Eubacterium eligens*, and Coprococcus [49]. Overall, SCFA supplementation has been shown to positively modulate GM composition across animal and human studies.

### 3.7. Monounsaturated Fatty Acids (MUFA)

MUFAs are unsaturated FAs containing a single double bond in their carbon chain [50]. MUFAs exist mainly in various plant sources, including olive oil, nuts, seeds, and avocados [51]. MUFAs mostly consist of palmitoleic acid (C16:1 n-7), oleic acid (C18:1 n-9), and eicosenoic acid (C20:1 n-9) [52]. Oleic acid, found abundantly in safflower, rapeseed, olive, and peanut, is considered the predominant representative [52,53]. In a study with mice fed an HFD with MUFAs (16.1% of fat), there was an increased the abundance of *Bacteroidetes* and *Bifidobacterium* from the *Actinobacteria phylum* after 19 weeks [54]. Conversely, there was a decrease in the levels of *Lactobacillus* from the *Firmicutes* phylum, along with reduced abundances of *Clostridial cluster XIVa* (*Firmicutes*) and *Enterobacteriales* (*Proteobacteria*) [54]. In a 16-week study, C57BL/6J mice were fed high-fat diets including olive oil (45% energy, primarily MUFA). The olive oil diet increased the population of *Bacteroidaceae* and *Bacteroides*, which are beneficial bacteria [37]. In a 4-week clinical study, researchers examined the different effects of MUFA-rich oils, including canola oil (36% energy), canola/DHA oil (39% energy), and canola oleic oil (44% energy) on the GM composition of participants at risk of metabolic syndrome (MetS). The results showed an increase in the abundance of beneficial bacteria, such as *Coprobacillus*, *Faecalibacterium*, *Lactobacillus*, *Robinsoniella*, and *Tepidimicrobium Fusibacter*, as well as *Turicibacter* within the *Firmicutes* phylum. Furthermore, there were increases in the abundance of Flexithrix, *Parabacteroides*, and *Prevotella* within the *Bacteroidetes phylum*, along with elevated levels of *Enterobacteriaceaes* within the *Proteobacteria* phylum. Conversely, the abundance of the potentially harmful bacteria *Isobaculum* within the *Firmicutes* phylum decreased [55]. In a study with individuals having MetS, four treatments were administered: (1) muffins and cookies with high-oleic acid canola oil (HOCO)-DHA (3 g/day); (2) muffins and cookies with barley flour, high molecular weight barley β-glucan (4.36 g/day), and a blend of oils as a control; (3) muffins and cookies with a combination of barley flour, barley β-glucan, and HOCO-DHA (50 g/day); and (4) muffins and cookies with all-purpose flour and a control oil (50 g/day). The study found no effects on the GM [56]. Similarly, in a randomized trial, the participants consumed three types of virgin olive oil for 3 weeks: (1) virgin olive oil (VOO, 80 mg PC/kg), (2) PC-enriched VOO (500 mg PC/kg, FVOO), and (3) PC-enriched VOO with thyme (500 mg PC/kg, FVOOT). FVOOT reduced oxidized LDL and increased *Bifidobacteria* and phenolic metabolites, suggesting cardiovascular and gut health benefits. FVOO raised coprostanone and fecal hydroxytyrosol but did not affect blood lipids or microbiota [57]. In addition, the evidence from a review study suggests that MUFA-rich diets positively modulate the composition of GM [58]. To summarize, MUFAs demonstrate the potential to promote beneficial bacteria while reducing harmful taxa.

### 3.8. Trans Fatty Acids (TFA)

TFAs are commonly found in two main sources: ruminant TFA (r-TFA) and industrial TFA (i-TFA) [59]. r-TFAs are present in foods derived from these animals, such as dairy products, meat, and other animal-based food sources. *Trans*-vaccenic acid (TVA, *trans*11–18:1) accounts for up to 70% of r-TFA and *trans*-palmitoleic acid (TPA, *trans*11–16:1) [59]. i-TFA is produced during the partial hydrogenation of vegetable oils used in food processing and is found in many processed foods, such as baked goods, fried foods, and snack foods [60]. Elaidic acid (EA; *trans*-18:1, n-9) is the major i-TFA in the food supply [59]. A mice study showed that the intake of i-TFA from soybean oil (23.60% of 1000 gr diet) increased the abundance of harmful bacteria such as Proteobacteria [61]. Further, the effects of low (4.27%) and high (23.60%) EA from hydrogenated soybean in the diet increased the abundance of beneficial *Lactobacillaceae* for 8 weeks in C57BL/6 mice. However, the high EA intake *demonstrated* adverse effects, including increased levels of Proteobacteria, *Desulfovibrionaceae*, and inflammatory parameters [61]. Furthermore, beneficial taxa, such as Bacteroidetes, Lachnospiraceae, and *Bacteroidales* S24-7, decreased in abundance, and there was a reduction in the fecal levels of butyric acid and valeric acid after high EA intake [61]. Finally, a recent mice study compared the effects of i-TFA and r-TFA (2–3% TPA or EA) intake on the GM. The results showed that TPA decreased the abundance of Staphylococcus sp55 but increased Staphylococcus sp119. On the other hand, EA increased the abundance of Staphylococcus sp119 but decreased *Ruminococcaceae UCG-014*, *Lachnospiraceae*, *and Clostridium sensu stricto 1* after 28 days [62]. Additionally, TPA intake increased fecal SCFAs, while EA intake decreased fecal SCFAs [62]. Moreover, TPA intake increased IPA, a metabolite derived from the tryptophan metabolism that decreases inflammation [62]. In sum, research shows that r-TFAs may lead to an increase in beneficial bacteria compared to i-TFA.

Conjugated linoleic acid (CLA) is a type of n-6 PUFA found in dairy products and beef [63]. CLA can also be synthesized by specific gut bacteria, such as *Lactobacillus*, *Butyrivibrio*, and *Megasphaera* [64,65]. In an 8-week mice study, the supplementation with t10c12-CLA (0.5%, *w*/*w*) led to lower proportions of *Firmicutes* and higher proportions of *Bacteroidetes* compared to the control group. Specifically, the increased levels of *Bacteroidetes*, in particular *Porphyromonadaceae* bacteria, may be associated with negative effects on lipid metabolism and the development of hepatic steatosis [66]. In a study on neonatal calves, maternal supplementation with CLA (38 g/day) or CLA combined with essential FA (CLA + EFA; CLA with linseed and safflower oils) was investigated. The CLA + EFA group showed a decrease in *Enterobacterales* (*Proteobacteria*) and an increase in *Chloroflexi* in the jejunal microbiota. Additionally, CLA + EFA supplementation increased the proportion of n-3 PUFAs in the jejunal chyme, highlighting its potential to positively modulate both fatty acid composition and gut microbiota [67]. Additionally, in an anaerobic batch culture model using human fecal microbiota, supplementation with linoleic acid (LA), bovine serum albumin (BSA), and inulin increased the production of CLA, particularly the cis9, *trans*11 isomer, at pH 5.5. This treatment enhanced the relative abundance of *Collinsella aerofaciens* and Bifidobacteria, which are linked to beneficial gut health effects, while reducing *Parabacteroides*, *Bilophila*, *Clostridia*, and *Enterobacteriaceae*, bacteria associated with adverse health outcomes [68]. CLA demonstrates context-dependent effects, with beneficial outcomes observed in specific dietary scenarios, such as increased n-3 PUFAs and improved microbial diversity.

### 3.9. Polyunsaturated Fatty Acids (PUFAs)

N-3 PUFAs include alpha-linolenic acid (ALA, C18:3 n-3), *eicosapentaenoic* acid (EPA, C20:5 n-3), and docosahexaenoic acid (DHA, C22:6 n-3), which are mainly found in flaxseeds, chia seeds, walnuts, and fatty fish. Animal studies have shown that n-3 PUFA intake can reduce the abundance of pro-inflammatory bacteria, such as *Fusobacterium nucleatum*, and enhance the growth of beneficial bacteria, such as *Bifidobacterium* and *Lactobacillus* species [69]. Moreover, the supplementation of an HFD with EPA and DHA (3000 mg/kg/d) increased *Firmicutes*, particularly the *Lactobacillus* group, when compared with the HFD (60% kcal) in a 19-week study on healthy mice [54]. In another animal study, an HFD with EPA and DHA (1 g per 100 g of diet) showed an increase in *Bifidobacteria* (*Actinobacteria*) and a decrease in *Firmicutes*, *Tenericutes*, and *Enterobacteria* (*Proteobacteria*) after 8 weeks [70]. Additionally, healthy men were provided with a vegetable-rich diet containing over 600 mg of n-3 PUFAs from fish oil sources for 2 weeks. The results demonstrate an increase in several beneficial Firmicutes genera, including *Blautia*, *Coprococcus*, *Ruminococcus*, *Subdoligranulum*, *Eubacterium*, *Anaerostipes*, and *Pseudobutyrivibrio*. However, the abundance of *Roseburia* and Faecalibacterium *prausnitzii* (*Firmicutes*) decreased. Additionally, *Akkermansia* spp. (*Verrucomicrobia*), *Bacteroidetes*, and *Actinobacteria* decrease following the intervention [71]. Further, n-3 PUFAs also serve as substrates to produce bacterial metabolites such as SCFAs [72]. In a study, participants receiving a daily 500 mg n-3 PUFA supplementation demonstrated increases in isobutyrate, isovalerate, and butyrate levels [73]. Moreover, the presence of n-3 PUFA derived from Mediterranean foods, as assessed by a food frequency questionnaire (FFQ), was associated with higher levels of propionate and butyrate in feces [74]. A study demonstrated that n-3 PUFA (20% kcal-5% kcal from EPA/DHA) can enhance the presence of NATs (taurine-conjugated amphipathic lipids) in bile. Specifically, C22:6 NAT can help reduce excessive intestinal lipid absorption [26]. Overall, animal and human studies with n-3 PUFAs increase beneficial bacteria and promote SCFA production.

N-6 PUFAs include linoleic acid (LA, C18:2 n-6), gamma-linolenic acid (GLA, C18:3 n-6), dihomo-gamma-linolenic acid (DGLA, C20:3 n-6), and arachidonic acid (AA, C20:4 n-6), which are found in vegetable oils and animal-derived foods [75]. A study with mice on an HFD (45% of diet) with high LA showed an increase in the *Clostridiaceae* (Firmicutes) and *Desulfovibrionaceae* (Proteobacteria) families and a decrease in the *Bacteroidaceae*, *Prevotellaceae*, and *Rikenellaceae* (Bacteroidetes) families [76]. Similarly, C57BL/6J mice fed safflower oil (rich in LA) for 8 weeks exhibited an increase in *Bacteroidetes* and *Clostridium clusters* XI, XVII, and XVIII (*Firmicutes*), along with a decrease in *Bacilli* (*Firmicutes*) [33]. In a 4-week crossover study, subjects at risk of MetS consumed an HFD (42% of the energy) with a blend of corn and safflower oil (n-6; 25/75) or a blend of n-3 PUFA flax and n-6 PUFA safflower oil (6/4) [55]. The results indicated an increase in the abundance of *Isobaculum*, which is typically beneficial. On the other hand, *Parabacteroides* and Prevotella, which are generally associated with positive gut health, decreased in abundance. Additionally, the abundance of Enterobacteriaceae and *Turicibacter* decreased, which are considered potentially harmful. Overall, the effects of n–6 PUFA are inconsistent. However, both animal and human studies report increases in beneficial *Firmicutes* (e.g., *Clostridiaceae* in animals, *Isobaculum* in humans) and decreases in *Bacteroidetes* (e.g., *Prevotellaceae* in animals, *Parabacteroides* in humans).

### 3.10. Sterols

Sterols do not contain any FAs but rather have a multiring structure and can be categorized into two main types: cholesterol and phytosterols [77]. Dietary cholesterol is found in animal-based foods, such as eggs, meat, poultry, and full-fat dairy products [78]. The American Heart Association (AHA) no longer recommends a specific numerical limit for dietary cholesterol [79]. However, the 2015–2020 edition of the Dietary Guidelines for Americans no longer includes a specific recommendation to limit dietary cholesterol intake to a maximum of 300 mg/d [80]. In a study, C57BL/6 mice received low (0.013%) and high (0.203%) cholesterol intake for 14 months. The GM levels of Mucispirillum, *Desulfovibrio*, *Anaerotruncus*, and *Desulfovibrionaceae* increased, while Bifidobacterium and Bacteroides were reduced in mice fed a high-fat high-cholesterol diet (HF-HCD) diet [81]. Furthermore, mice received a control diet (10% fat), HFD (60% fat), and high-fat high-cholesterol diet (40% fat + 1.25% cholesterol) for 12 weeks. The results show that the HF- HCD decreased *Bacteroidetes* and *Firmicutes* while increasing *Proteobacteria*. Additionally, there was an increase in the *Firmicutes* and *Bacteroidetes* ratio in a high-fat high-cholesterol diet compared to both the normal and HFD groups [82]. In another study, mice who received a 1.25% cholesterol diet for 12 weeks showed no differences in either the total or relative abundance of the primary microbial phyla in the cecum [83]. Overall, high dietary cholesterol may cause imbalances in the GM.

Further, phytosterols are a group of sterols that are naturally found in various plant-based foods, such as nuts, seeds, and vegetable oils. In a C57BL/6J mice study, three groups were established: a normal diet group, an HFD group, and a high-fat diet with phytosterol (100 mg/kg) group. Phytosterol treatment resulted in decreased *Lactobacillus* and bile salt hydrolase (BSH) production by GM in HFD-fed mice [84]. In another study, the administration of 87.8 mg/kg/day of three phytosterols (daucosterol linolenate (DLA), daucosterol linoleate (DL), and daucosterol palmitate (DP)) for 29 days in mice resulted in an increase in Bacteroidetes and a decrease in Firmicutes. In addition, daucosterol linoleate promoted butyric acid production, and daucosterol palmitate promoted both acetic acid and butyric acid production in the colorectal and feces [85]. In a study on mice, the administration of 300 and 600 mg/kg of phytosterol extract for 5 weeks resulted in reductions in the *Firmicutes*/*Bacteroidetes* ratio, *Firmicutes*, *Proteobacteria*, and *Verrucomicrobia* abundances, and fecal bile acid, along with notable increases in *Bacteroidetes* and *Actinobacteria* abundances [86]. Yet, in a human study with a supplementation of 3 g/d of plant stanol ester for 3 weeks, a change in the composition or diversity of microbiota was not experienced [87]. Overall, studies suggest that phytosterols may positively influence the GM composition in animal studies, but this is not replicated in human studies (Table 1, Figure 1).

## 4. Limitations and Conclusions

First, differences in microbiome composition between species influence how dietary interventions affect gut health. Mice are commonly used as models due to their controlled environments and simplified diets. However, the GMs of animals differ from humans, leading to potentially divergent responses. Additionally, human studies often involve more complex dietary interventions, including blends of lipids and varied food sources, whereas animal studies typically utilize isolated fat sources, making direct comparisons challenging. Furthermore, the dose of lipids used in animal studies is often higher relative to body weight compared to humans, which may complicate the translation of findings to human populations. The length of dietary interventions also varies widely, with animal studies typically lasting 3 to 21 weeks and human studies spanning 2 to 24 weeks. However, the optimal duration to induce changes in the gut microbiota remains uncertain and may depend on factors such as the type of lipid, the dosage, and the baseline microbiome composition. Additionally, the dietary context, including the fiber content and overall macronutrient composition, can interact with fat intake to shape gut microbiota outcomes. A key limitation of animal studies is that HFDs often contain lower fiber compared to standard chow diets [84], potentially confounding their effects on the GM. As fiber substantially shapes microbiota composition and SCFA production, it is difficult to attribute changes solely to dietary lipid intake. Additionally, a direct comparison of SFAs with different chain lengths, particularly VLCSFA (20 carbon atoms and more), is needed to understand their distinct impacts on the gut microbiome and metabolism. Further, the potential synergistic or additive effects of combining different types of dietary lipids on the GM and its metabolites remain largely unexplored. The baseline health status of study populations is also important. Animals are often studied under healthy or induced-disease conditions, while human participants frequently have pre-existing metabolic risk factors. Finally, studies should also measure both the GM composition and the key gut metabolites before and after dietary interventions to comprehensively evaluate their effects on gut health. In sum, these gaps emphasize the need for well-designed, standardized human studies that account for dietary complexity, fiber intake, and baseline health status to better understand the effects of dietary lipids on the GM and their translation from animal models.

Up to now, the data demonstrate that the intake of an HFD that includes long-chain SFAs, industrial TFAs, and cholesterol is associated with detrimental changes in GM composition and reduced SCFA production, while the intake of MUFAs, n-3 PUFAs, and short-chain FAs may increase beneficial bacteria and metabolites. Nevertheless, additional human clinical trials are needed to understand and adjust dietary fat recommendations for optimal gut health.

## Figures and Tables

**Figure 1 nutrients-17-00639-f001:**
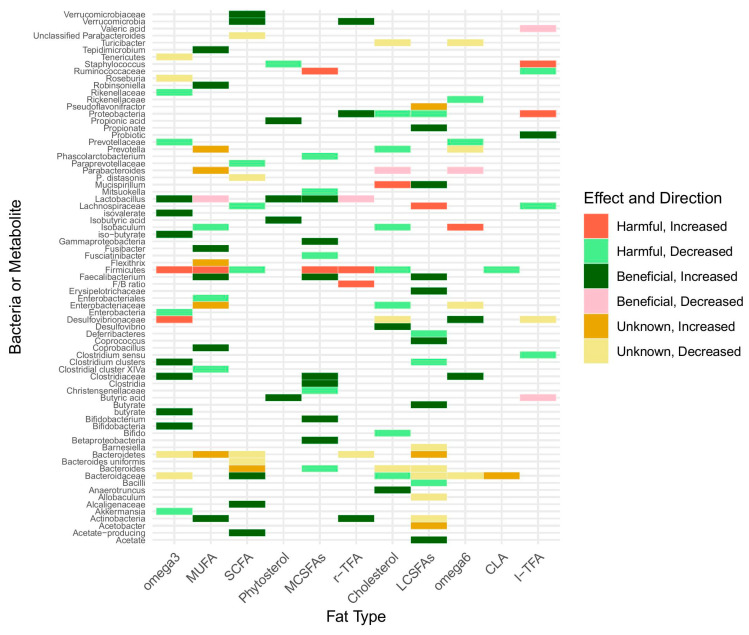
Heatmap illustrating the impact of various dietary lipids, including fatty acids and sterols on gut microbiota and their metabolites. Rows represent bacterial taxa or metabolites, and columns represent fatty acids and sterol types. Colors indicate the direction (increase or decrease) and relevance (beneficial, harmful, or unknown).

**Table 1 nutrients-17-00639-t001:** Change in microbiota and metabolites composition after dietary lipids intake.

Study Type	Intervention	Change in Microbiomes	Change in Metabolites and MetS
**n-3 PUFA**			
Animal (Mice) [54]	HFD with EPA and DHA (3000 mg/kg/d), 19 weeks	↑ *Firmicutes*, *particularly Lactobacillus*	NM
Animal (Mice) [70]	HFD with EPA and DHA (1 g per 100 g of diet), 8 weeks	↑ *Bifidobacteria* (*Actinobacteria*), ↓ *Firmicutes*, ↓ *Tenericutes*, ↓ *Enterobacteria* (*Proteobacteria*)	NM
Animal (Mice) [76]	HFD rich in LA for 21 weeks	↑ *Clostridiaceae* (*Firmicutes*), ↑ *Desulfovibrionaceae* (*Proteobacteria*); ↓ *Bacteriodaceae*, ↓ *Prevotellaceae*, ↓ *Rickenellaceae* (*Bacteroidetes*)	NM
Animal (Mice) [33]	Safflower oil (rich in LA; 177.5 gr) for 8 weeks	↑ *Bacteroidetes*, ↑ *Clostridium cluster* XI, XVII, and XVIII (*Firmicutes*); ↓ *Bacilli* (*Firmicutes*)	NM
Human [71]	Vegetable-rich diet with over 600 mg of n-3 PUFAs from fish oil sources, 2 weeks	↑ *Beneficial Firmicutes genera* (*Blautia*, *Coprococcus*, *Ruminococcus*, *Subdoligranulum*, *Eubacterium*, *Anaerostipes*, *Pseudobutyrivibrio*); ↓ *Roseburia*, ↓ *Faecalibacterium prausnitzii* (*Firmicutes*); ↓ *Akkermansia* spp. (*Verrucomicrobia*), ↓ *Bacteroidetes*, ↓ *Actinobacteria*	NM
Human [73]	500 mg n-3 supplementation daily, 6 weeks	NM	↑ iso-butyrate, isovalerate, and butyrate levels
Human [74]	n-3 FA from Mediterranean foods (nuts and fatty fish), assessed by FFQ	NM	↑ propionate and butyrate in feces; Improved intestinal barrier integrity
Human [26]	n-3 FA (20% kcal-5% kcal from EPA/DHA), enhancing NATs (taurine-conjugated amphipathic lipids) in bile, 6 weeks	NM	Enhanced presence of NATs, specifically C22:6 NAT; Contributing to hypotriglyceridemic mechanism
Human [55]	HFD with blend of corn and safflower oil (n-6; 25/75) or blend of n-3 PUFA flax and n-6 PUFA safflower oil (6/4), 4 weeks	↑ *Isobaculum* (*Firmicutes*); ↓ *Parabacteroides*, ↓ *Prevotella* (*Bacteroidetes*); ↓ *Enterobacteriaceae*, ↓ *Turicibacter* (*Firmicutes*)	NM
**Short-Chain FA**			
Animal (Mice) [48]	HFD + Sodium butyrate (400 mg/kg), 16 weeks	↑ *Verrucomicrobia*, ↓ *Firmicutes*; ↑ *Verrucomicrobiaceae*, ↑ *Bacteroidaceae*, ↑ *Alcaligenaceae*, ↓ *Lachnospiraceae*, ↓ *Paraprevotellaceae*	↔ total cholesterol and triglyceride levels
Animal (Mice) [47]	Butyrate supplementation (0.5 g/kg/day), 7 weeks	↑ *Bacteroidetes*, ↓ *Firmicutes/Bacteroidetes ratio*;	↓ blood glucose and animal body weight
Animal (Mice) [46]	Acetate supplementation (200 mmol/L magnesium acetate), 3 weeks	↓ *Firmicutes*: *Bacteroidetes ratio*; ↑ *Acetate-producing bacteria*, ↑ *Bacteroides*	↓ systolic BP ↓ body weight
Human [49]	Acetate and Butyrate supplements (40 g/day), 6 weeks	↑ *Bacteroides uniformis*, ↑ *Unclassified Parabacteroides*, ↑ *P. distasonis*; ↓ *Eubacterium ramulus*, ↓ *Eubacterium eligens*, ↓ *Coprococcus*;	↓ HbA1c **↓** Plasma IL-8, ↓ MIP-1α, ↓ bFGF
**MUFA**			
Animal (Mice) [54]	HFD with MUFA (16.1% of fat), 19 weeks	↑ *Bacteroidetes*, ↑ *Bifidobacterium* (*Actinobacteria*); ↓ *Lactobacillus* (*Firmicutes*), ↓ *Clostridial cluster* XIV*a* (*Firmicutes*), ↓ *Enterobacteriales* (*Proteobacteria*)	NM
Human [55]	MUFA-rich oils (Canola oil, Canola/DHA oil, Canola oleic oil), 4 weeks	↑ *Coprobacillus*, ↑ *Faecalibacterium*, ↑ *Lactobacillus*, ↑ *Robinsoniella*, ↑ *Tepidimicrobium Fusibacter* (*Firmicutes*); ↑ *Flexithrix*, ↑ *Parabacteroides*, ↑ *Prevotella* (*Bacteroidetes*); ↑ *Enterobacteriaceaes* (*Proteobacteria*); ↓ *Isobaculum* (*Firmicutes*)	NM
Human [56]	Treatments: (1) HOCO-DHA (3 g/day); (2) Control with barley flour and barley β-glucan (4.36 g/day); (3) Combination of barley flour, barley β-glucan, and HOCO-DHA (50 g/day); (4) Control with all-purpose flour and control oil (50 g/day)	↔ microbiomes	↓ LDL cholesterol levels
**r-TFA vs. i-TFA**			
Animal (Mice) [62]	i-TFA and r-TFA (2–3% TPA or EA) intake, 7 and 28 days	TPA: ↓ *Staphylococcus* sp*55*, ↑ *Staphylococcus* sp*119*; EA: ↑ *Staphylococcus* sp*119*, ↓ *Ruminococcaceae UCG-014*, ↓ *Lachnospiraceae*, ↓ *Clostridium sensu stricto*	TPA: ↑ Butyric, ↑ Isobutyric, ↑ Propionic acids; EA: ↓ Isobutyric, ↓ Isocaproic, ↓ Isovaleric, ↓ Caproic, ↓ Valeric, ↓ Heptanoic acids; TPA: ↑ IPA; EA: ↓ IPA
Animal (Mice) [66]	t10c12-CLA supplementation (0.5%, *w*/*w*), 8 weeks	↓ *Firmicutes*, ↑ *Bacteroidetes*, ↑ *Porphyromonadaceae*	Negative effects on lipid metabolism, development of hepatic steatosis
Animal (Calves) [67]	CLA (38 g/day) and CLA + EFA (CLA + linseed + safflower oils)	CLA + EFA decreased *Enterobacterales* (*Proteobacteria*) and increased *Chloroflexi*.	CLA + EFA increased n-3 FAs in jejunal chyme
**Medium-chain SFA**			
Animal (Broiler Chickens) [42]	Lauric acid supplementation (500 mg/kg and 1000 mg/kg), 42 days	↓ *Phascolarctobacterium*, ↓ *Christensenellaceae_R-7_group*, ↓ *Bacteroides*, ↑ *Faecalibacterium*, ↑ *Ruminococcaceae_UCG-014*	↑ Body weight, ↑ Average daily gain, ↓ IL-1b, ↓ IL-6, ↓ TNF-a, ↓ IL-4, ↓ IL-10
Animal (Black Sea Bream) [41]	Diets with 0.1% and 0.8% lauric acid, 8 weeks	↑ *Firmicutes*, ↑ *Betaproteobacteri*, ↑ *Gammaproteobacteria*, ↑ *Clostridia*, ↑ *Clostridiaceae*	NM
Animal (Pigs) [40]	Commercial coconut oil supplementation (0.3%), 14 days	↑ *Lactobacillus*, ↑ *Bifidobacterium*, ↓ *Fusicatenibacter*, ↓ *Mitsuokella*	NM
Human (Overweight Women) [43]	Coconut oil consumption (25 mL/day) as part of an energy-restricted diet, 9 weeks	↔ gut microbial diversity	NM
**n-6 PUFA**			
Animal (Mice) [76]	HFD rich in LA for 21 weeks	↑ *Clostridiaceae* (*Firmicutes*), ↑ *Desulfovibrionaceae* (*Proteobacteria*), ↓ *Bacteriodaceae*, ↓ *Prevotellaceae*, ↓ *Rickenellaceae* (*Bacteroidetes*)	NM
Animal (Mice) [33]	Safflower oil (rich in LA; 177.5 gr) for 8 weeks	↑ *Bacteroidetes*, ↑ *Clostridium cluster* XI, XVII, XVIII (*Firmicutes*), ↓ *Bacilli* (*Firmicutes*)	NM
Animal (Mice) [37]	Olive oil (45% energy, high MUFA) for 16 weeks	↑ *Bacteroidaceae*, ↑ *Bacteroides* (*beneficial bacteria*), ↓ *harmful bacteria compared to SFA or n-3 PUFA diets*	Reduced fat mass, improved metabolic markers
Human (Subjects at risk of MetS) [55]	HFD with a blend of corn and safflower oil (n-6; 25/75) or blend of n-3 PUFA flax and n-6 PUFA safflower oil (6/4), 4 weeks	↑ *Isobaculum* (*Firmicutes*); ↓ *Parabacteroides*, ↓ *Prevotella* (*Bacteroidetes*); ↓ *Enterobacteriaceae*, ↓ *Turicibacter* (*Firmicutes*)	NM
**Long-chain SFA**			
Animal (Mice) [33]	Palm oil enriched HFD (10% kcal palmitic acid), 3 weeks	↑ *Bacteroidetes*, ↓ *Bacilli*, ↓ *Clostridium clusters* XI, XVII, XVIII (*Firmicutes*)	↑ Acetate and Propionate, ↓ Butyrate
Animal (Mice) [37]	High-fat palm oil-rich HFD (45% palmitic acid), 16 weeks	↑ *Coprococcus*, ↑ *Erysipelotrichaceae*, ↑ *Lachnospiraceae* (*Firmicutes*), ↓ *Bacteroides*, ↓ *Bacteroidaceae* (*Bacteroidetes*), ↓ *Deferribacteres*, ↓ *Actinobacteria*, ↓ *Proteobacteria*	NM
Animal (Mice) [22]	HFD with 34% SFAs, 8 weeks	↑ *Actobacillus*, ↑ *Erysipelotrichaceae*, ↑ *Lachnospiraceae*, ↑ *Pseudoflavonifractor* (*Firmicutes*), ↓ *Allobaculum* (*Firmicutes*), ↓ *Bamesiella* (*Bacteroidetes*), ↓ *Mucispirillum* (*Deferribacteres*), ↓ *Bacteroides* (*Bacteroidetes*), ↓ *Bifidobacterium* (*Actinobacteria*)	NM
Human [88]	High SFA diet (38% total fat, 18% from SFAs), 24 weeks	↑ *Faecalibacterium prausnitzii*	↓ Inflammation ↑ SCFA production, particularly butyrate
**i-TFA**			
Animal (Mice) [61]	Low (4.27%) and high (23.60%) EA from hydrogenated soybean, 8 weeks	↑ *Lactobacillaceae*; ↑ *Proteobacteria*, ↑ *Desulfovibrionaceae*; ↓ *Bacteroidetes*, ↓ *Lachnospiraceae*, ↓ *Bacteroidales S24-7*;	↑ IL-6 and TNF-α ↓ in fecal levels of butyric acid and valeric acid after high EA intake
**Cholesterol**			
Animal (Mice) [81]	Low (0.013%) and high (0.203%) cholesterol intake for 14 months	↑ *Mucispirillum*, ↑ *Desulfovibrio*, ↑ *Anaerotruncus*, ↑ *Desulfovibrionaceae*; ↓ *Bifidobacterium*, ↓ *Bacteroides*	NM
Animal (Mice) [82]	Control diet (10% fat), HFD (60% fat), HF-HC diet (40% fat + 1.25% cholesterol), 12 weeks	HF-HC diet: ↓ *Bacteroidetes*, ↓ *Firmicutes*, ↑ *Proteobacteria*; Changes in *Firmicutes*/*Bacteroidetes* ratio	↑ Dyslipidemia, insulin resistance, obesity
Animal (Mice) [83]	1.25%-cholesterol diet for 12 weeks	No differences in total or relative abundance of primary microbial phyla	
**Phytosterol**			
Animal (Mice) [84]	High-fat diet with phytosterol (100 mg/kg), C57BL/6J male mice	↓ *Lactobacillus*	↓ Total cholesterol, ↓ TG, ↓ LDL, ↑ HDL-C
Animal (Mice) [85]	Phytosterols (daucosterol linolenate, daucosterol linoleate, daucosterol palmitate, 87.8 mg/kg/day), 29 days	↑ *Bacteroidetes*, ↓ *Firmicutes*	↓ acetic acid and butyric acid ↑ MetS
Animal (Mice) [86]	Lotus seed core powder phytosterol extract (300 and 600 mg/kg), 5 weeks	↓ *F/B* ratio, ↓ *Firmicutes*, ↓ *Proteobacteria*, ↓ *Verrucomicrobia*,↑ *Bacteroidetes*, ↑ *Actinobacteria*	Potentially improve MetS-related factors
Human (Participants) [87]	Plant stanol ester (3 g/d), 3 weeks + 4-week wash-out	↔ microbiota composition or diversity	NM

HFD: high-fat diet; EPA: eicosapentaenoic acid; DHA: docosahexaenoic acid; NM: not mentioned; LA: linoleic acid; PUFA: polyunsaturated fatty acids; FFQ: food frequency questionnaire; SFA: saturated fatty acids; MUFA: monounsaturated fatty acids; MetS: metabolic syndrome; NAT: N-Acyl taurine; TPA: trans palmitoleic acid; EA: elaidic acid; IPA: isopropyl alcohol; SCFA: short-chain fatty acids; LDL: low-density lipoprotein; HDL-C: high-density lipoprotein cholesterol; TG: triglycerides; IL: interleukin; IL-1b: interleukin-1 beta; IL-6: interleukin-6; TNF: tumor necrosis factor; TNF-a: tumor necrosis factor-alpha; BP: blood pressure; HbA1c: hemoglobin A1c; bFGF: basic fibroblast growth factor; EFA: essential fatty acids; HF-HC: high-fat high cholesterol. ↑: increase; ↓ decrease.

## Data Availability

No new data were created or analyzed in this study. Data sharing is not applicable to this article as it is a review.

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
