# Peer review of "Dietary Lipids, Gut Microbiota, and Their Metabolites: Insights from Recent Studies"

_nutrients, 2025, doi:10.3390/nu17040639_

Round 1

Reviewer 1 Report

Comments and Suggestions for Authors

  1. The author did not separate dietary fat from fatty acids, and dietary fat should not include short chain fatty acids, which are classified according to the length of the fatty acid chain. In nutrition discussions, the term 'dietary fatty acids' typically refers to fatty acids that can be directly obtained from daily diet, especially those long-chain fatty acids that are crucial for human health. And the sources of short chain fatty acids summarized in the article are also fermented foods and beverages, and fatty acids and dietary fats cannot be discussed separately.

       2.In the description under the "Long chain SFA" item on line 144, high-fat diet and saturated fatty acid diet are both co causative and influencing factors, and can also be considered as pathogenic and influencing factors respectively. Suggest increasing the number of cases with saturated fatty acids as a separate pathogenic factor and influencing factor, and reducing cases with common pathogenic factors.

       3. "3. Food" in line 75 needs to be further subdivided. For example, the 132nd line "SFA" and the 144th line "long chain SFA" should be a relationship between the size of the titles, not a parallel relationship.

       4.It is recommended to add corresponding health effects after describing the effects of fatty acids on gut products or gut microbes.

       5.Sterols are also not dietary fats. Dietary fat mainly refers to triglycerides, which are triacylglycerides composed of glycerol and fatty acids, while sterols are a class of steroids, which contain hydroxyl groups. They are all based on cyclopentanpolyhydrophenanthrene and contain hydroxyl groups, so they are called sterols. Both are lipids, suggesting the authors change dietary fat to the more correct term. 

     6.The article has conceptual errors. It is suggested to use the correct words.The author needs to distinguish between the concepts of dietary fat and fatty acids, which cannot be discussed together. In nutrition discussions, the term "dietary fatty acids" usually refers to fatty acids that can be directly obtained from daily diet, especially those long-chain fatty acids that are crucial for human health. Dietary fats should not include short chain fatty acids, which are classified according to their chain length.Sterols are also not considered dietary fats.  Dietary lipids primarily refer to triglycerides—combinations of glycerol and three fatty acids—while sterols are steroid compounds with a hydroxyl group, based on a cyclopentanopolysilane structure.  

7.the article focuses on dietary fat, a topic that makes a lot of sense in nutrition. Some of the content is logically coherent, such as the discussion of the relationship between saturated fatty acids and other related factors. The reference of data is reliable and scientific, and it focuses on the cutting-edge topics of the impact of fatty acids on gut microorganisms, which is forward-looking.On the other hand, the article has many shortcomings. The concept of dietary fat and fatty acids is confused, food classification is not detailed, fatty acids affect gut microbes and lack of health effects, and sterols are mistaken as dietary fats, all of which affect the accuracy and completeness of the article. 

Author Response

Comments and Suggestions for Authors (Reviewer 1)

 The article focuses on dietary fat, a topic that makes a lot of sense in nutrition. Some of the content is logically coherent, such as the discussion of the relationship between saturated fatty acids and other related factors. The reference of data is reliable and scientific, and it focuses on the cutting-edge topics of the impact of fatty acids on gut microorganisms, which is forward-looking. On the other hand, the article has many shortcomings. The concept of dietary fat and fatty acids is confused, food classification is not detailed, fatty acids affect gut microbes and lack of health effects, and sterols are mistaken as dietary fats, all of which affect the accuracy and completeness of the article. 

  1. The author did not separate dietary fat from fatty acids, and dietary fat should not include short chain fatty acids, which are classified according to the length of the fatty acid chain. In nutrition discussions, the term 'dietary fatty acids' typically refers to fatty acids that can be directly obtained from daily diet, especially those long-chain fatty acids that are crucial for human health. And the sources of short chain fatty acids summarized in the article are also fermented foods and beverages, and fatty acids and dietary fats cannot be discussed separately.

Response: We thank the reviewer for highlighting the importance of distinguishing dietary fats, dietary fatty acids, and short-chain fatty acids (SCFAs) in our manuscript. To address this concern, we have made the following modifications:

(1) We carefully reviewed the manuscript, revised the text to use the term "fatty acid" where appropriate to ensure clarity and consistency in distinguishing dietary lipids from fatty acids and SCFAs, and updated the title accordingly to reflect this distinction.

(2) To further clarify the inclusion of dietary SCFAs in our review, we added the following sentence to the introduction (lines 60–62): Moreover, SCFAs can also be consumed directly through dietary sources such as fermented foods, beverages, and specific supplements which further improve gut health.

2. In the description under the "Long chain SFA" item on line 144, high-fat diet and saturated fatty acid diet are both co causative and influencing factors and can also be considered as pathogenic and influencing factors respectively. Suggest increasing the number of cases with saturated fatty acids as a separate pathogenic factor and influencing factor and reducing cases with common pathogenic factors.

          Response We appreciate the reviewer’s feedback and acknowledge that there might be some confusion due to the concise descriptions of the studies in the manuscript. To clarify, in reference [22], the high saturated fat (SFA) diet was compared to a high-fat diet enriched in n-6 fatty acids (HFD-n6). Similarly, in reference [36], the high saturated fat diet was compared to diets containing (1) palm oil, (2) olive oil, (3) safflower oil, or (4) flaxseed/fish oil. These differences in study designs and control diets mean that there are no direct causal conclusions or consistent influencing factors for the effects of high-fat diets and saturated fat across the studies. To address this, we have modified the text for these two studies to improve clarity.  Lines 159-166 and 171-175

In another study, C57BL/6J mice received both a HFD containing 34% saturated fatty acids (SFAs; type of fat not specified) and an HFD enriched with omega-6 fatty acids (HFD-n6). The HFD with SFA increased the abundance of Lactobacillus, Erysipelotrichaceae, Lachnospiraceae, and Pseudoflavonifractor (Firmicutes) after 8 weeks. In contrast, Allobaculum (Firmicutes), Barnesiella (Bacteroidetes), Mucispirillum (Deferribacteres), Bacteroides (Bacteroidetes), and Bifidobacterium (Actinobacteria) exhibited a decrease in abundance following the SFA-enriched HFD [22].

And

Similarly, a HFD rich in 45% of either palm oil, olive oil, safflower oil, or flaxseed/fish oil was administered to C57BL/6J mice for 16 weeks. The palm oil-enriched diet increased the abundance of Coprococcus, Erysipelotrichaceae, and Lachnospiraceae (Firmicutes) [36]. Conversely, the abundance of Bacteroides and Bacteroidaceae (Bacteroidetes), as well as Deferribacteres, Actinobacteria, and Proteobacteria, decreased after receiving palm oil [36].

  1. Food" in line 75 needs to be further subdivided. For example, the 132nd line "SFA" and the 144th line "long chain SFA" should be a relationship between the size of the titles, not a parallel relationship.

Response: We acknowledge the reviewer’s observation regarding the organization of the section and the relationship between "SFA" and "long-chain SFA". The format was adjusted by the editor during the revision process. We will follow up with the editor to ensure that this comment is appropriately addressed in the final version.

  1. It is recommended to add corresponding health effects after describing the effects of fatty acids on gut products or gut microbes.

Response: Thank you for the suggestion. We had previously included the corresponding health effects in the manuscript but later shortened this part to streamline the paper, as the primary objective focuses on microbiomes and metabolites. However, we agree that this could be a valuable idea for another study and appreciate the recommendation.

  1. Sterols are also not dietary fats. Dietary fat mainly refers to triglycerides, which are triacylglycerides composed of glycerol and fatty acids, while sterols are a class of steroids, which contain hydroxyl groups. They are all based on cyclopentanpolyhydrophenanthrene and contain hydroxyl groups, so they are called sterols. Both are lipids, suggesting the authors change dietary fat to the more correct term. 

Response: Thank you for pointing this out. We have revised the text to clarify that sterols are a distinct class of lipids and not part of dietary fat, which primarily refers to triglycerides. To do so, we updated the sterol section and ensured the distinction between sterols and triglycerides is explicitly stated in the text.  Lines 345-347:

Sterols do not contain any fatty acids but rather have a multiring structure and can be categorized into two main types: cholesterol and phytosterols.

  1. The article has conceptual errors. It is suggested to use the correct words. The author needs to distinguish between the concepts of dietary fat and fatty acids, which cannot be discussed together. In nutrition discussions, the term "dietary fatty acids" usually refers to fatty acids that can be directly obtained from daily diet, especially those long-chain fatty acids that are crucial for human health. Dietary fats should not include short chain fatty acids, which are classified according to their chain length. Sterols are also not considered dietary fats.  Dietary lipids primarily refer to triglycerides—combinations of glycerol and three fatty acids—while sterols are steroid compounds with a hydroxyl group, based on a cyclopentanopolysilane structure.  

Response: Thank you for your observation. As we responded in comments 1 and 5:
A: We carefully reviewed the manuscript and revised the text to use the term "fatty acid" where appropriate to ensure clarity and consistency in distinguishing dietary lipids from fatty acids and SCFAs.

 B: We have revised the text to clarify that sterols are a distinct class of lipids and not part of dietary fat. These changes ensure scientific accuracy and address the conceptual distinctions highlighted in your comment.

Sterols do not contain any fatty acids but rather have a multiring structure and can be categorized into two main types: cholesterol and phytosterols.

Reviewer 2 Report

Comments and Suggestions for Authors

General comments:

the manuscript is a non-systematic review of trials (mainly in animals) in which the influence of specific types of fat on the intestinal microbiota and on some metabolites, mainly short-chain fatty acids, has been investigated. Overall, it cannot be said to be a very novel review, but it is generally well written and may be of some interest to the reader. Some aspects that need to be improved are mentioned as specific comments

Specific comments:

Line 31. “trans” and “omega” should be written in italics

Lines 53-54 “The main SCFAs generated by gut bacteria are acetate,  propionate, and butyrate”. Depending of the gut microbiota composition, lactate can be also between the most produced SCFAs.

Lines 66-74: More information should be provided about the criteria, keywords used in the search, exclusion of articles, etc., and cite the search and selection algorithm in detail.

Line 79: “HFD” must be defined the its time that appears in the text.

Line 80: Bacterial genus names must be written in italics. The same for all the rest of the manuscript.

Line 132: Subheadings should not be abbreviations

Table 1: “change in microbiomes” should be better named “change in microbiota”.

There are abbreviation in Table 1 that are not defined in the Table footnote, such as “HF-HC”,”NAT”, etc. Additionally, it would be better if abbreviations are defined in alphabetical order.

Figure 1 needs a much more detailed explanation, at least about the most relevant findings.

References list is not in the font type, size and format recommended by MDPI in its instructions for authors.

Author Response

 (Reviewer 2)

General comments:

the manuscript is a non-systematic review of trials (mainly in animals) in which the influence of specific types of fat on the intestinal microbiota and on some metabolites, mainly short-chain fatty acids, has been investigated. Overall, it cannot be said to be a very novel review, but it is generally well written and may be of some interest to the reader. Some aspects that need to be improved are mentioned as specific comments

Specific comments:

  1. Line 31. “trans” and “omega” should be written in italics

Response: Thank you for your comment. We have revised the manuscript to ensure that “trans” and “omega” are written in italics as suggested.

  1. Lines 53-54 “The main SCFAs generated by gut bacteria are acetate, propionate, and butyrate”. Depending of the gut microbiota composition, lactate can be also between the most produced SCFAs.

Response: Thank you for your comment. We would like to clarify that lactate is not technically classified as a short-chain fatty acid (SCFA).

  1. Lines 66-74: More information should be provided about the criteria, keywords used in the search, exclusion of articles, etc., and cite the search and selection algorithm in detail.

Response: Thank you for your feedback. We have expanded the methods section to provide additional details about the criteria, keywords, and article selection process. Lines 69-83:

This narrative review synthesizes existing literature to explore the impacts of die-tary fats, fatty acids and sterols on GM and metabolites. Studies were identified through a systematic search of the PubMed, Medline, Scopus, and Google Scholar da-tabases up to April 2024. The search included the following keywords: “dietary fatty acids,” “dietary fat,” “gut microbiota,” “short-chain fatty acids,” “metabolic syn-drome,” “gut metabolites,” “long-chain fatty acids,” “saturated fatty acids,” “mono-unsaturated fatty acids,” “polyunsaturated fatty acids,” “omega-3 fatty acids,” “ome-ga-6 fatty acids,” “trans fatty acids,” “gut dysbiosis,” “microbial diversity,” “gut in-flammation,” “host-microbiome interactions,” “cholesterol,” “dietary cholesterol,” “sterols,” “phytosterols,” and “plant sterols.”" The inclusion criteria encompassed clinical and pre-clinical studies focused on dietary fats, including FAs and sterols and their effects on GM and metabolites, providing a comprehensive overview for this explanatory review. Articles not published in English or lacking full-text availability were excluded. The screening process involved an initial review of titles and abstracts, followed by a full-text review for eligible studies. A total of 33 studies published up to 2024 were selected for inclusion.

  1.  
  2. Line 79: “HFD” must be defined its time that appears in the text.

Response: Corrected

  1. Line 80: Bacterial genus names must be written in italics. The same for all the rest of the manuscript.

Response: Corrected

  1. Line 132: Subheadings should not be abbreviations

Response: Thank you for pointing this out. We have updated the subheadings.

  1. Table 1: “change in microbiomes” should be better named “change in microbiota”.

Response: Corrected

  1. There are abbreviation in Table 1 that are not defined in the Table footnote, such as “HF-HC”,”NAT”, etc. Additionally, it would be better if abbreviations are defined in alphabetical order.

  Response: Thank you for your comment. We have added the missing abbreviations to the footnote of Table 1. Additionally, we have reordered all abbreviations to follow their order of appearance in the table rather than alphabetical order for improved clarity and consistency.

  1. Figure 1 needs a much more detailed explanation, at least about the most relevant findings.

 Response: Added
